# Substitution of Coxsackievirus A16 VP1 BC and EF Loop Altered the Protective Immune Responses in Chimera Enterovirus A71

**DOI:** 10.3390/vaccines11081363

**Published:** 2023-08-14

**Authors:** Xiu Hui Tan, Wei Lim Chong, Vannajan Sanghiran Lee, Syahril Abdullah, Kartini Jasni, Saiful Qushairi Suarni, David Perera, I-Ching Sam, Yoke Fun Chan

**Affiliations:** 1Department of Medical Microbiology, Faculty of Medicine, Universiti Malaya, Kuala Lumpur 50603, Malaysia; mmf190017@siswa.um.edu.my (X.H.T.); jicsam@um.edu.my (I.-C.S.); 2Department of Chemistry, Center of Theoretical and Computational Physics, Faculty of Science, Universiti Malaya, Kuala Lumpur 50603, Malaysia; 3Department of Biomedical Sciences, Faculty of Medicine & Health Sciences, Universiti Putra Malaysia, Serdang 43400, Malaysia; syahril@upm.edu.my; 4Comparative Medicine and Technology Unit, Institute of Bioscience, Universiti Putra Malaysia, Serdang 43400, Malaysia; 5Institute of Health and Community Medicine, Universiti Malaysia Sarawak, Kota Samarahan 94300, Malaysia; dperera@unimas.my

**Keywords:** enterovirus A71, coxsackievirus A16, enterovirus, hand, foot and mouth disease, capsid loop, vaccine, immunogenicity

## Abstract

Hand, foot and mouth disease (HFMD) is a childhood disease caused by enterovirus A71 (EV-A71) and coxsackievirus A16 (CV-A16). Capsid loops are important epitopes for EV-A71 and CV-A16. Seven chimeric EV-A71 (ChiE71) involving VP1 BC (45.5% similarity), DE, EF, GH and HI loops, VP2 EF loop and VP3 GH loop (91.3% similarity) were substituted with corresponding CV-A16 loops. Only ChiE71-1-BC, ChiE71-1-EF, ChiE71-1-GH and ChiE71-3-GH were viable. EV-A71 and CV-A16 antiserum neutralized ChiE71-1-BC and ChiE71-1-EF. Mice immunized with inactivated ChiE71 elicited high IgG, IFN-γ, IL-2, IL-4 and IL-10. Neonatal mice receiving passive transfer of WT EV-A71, ChiE71-1-EF and ChiE71-1-BC immune sera had 100%, 80.0% and no survival, respectively, against lethal challenges with EV-A71, suggesting that the substituted CV-A16 loops disrupted EV-A71 immunogenicity. Passive transfer of CV-A16, ChiE71-1-EF and ChiE71-1-BC immune sera provided 40.0%, 20.0% and 42.9% survival, respectively, against CV-A16. One-day-old neonatal mice immunized with WT EV-A71, ChiE71-1-BC, ChiE71-1-EF and CV-A16 achieved 62.5%, 60.0%, 57.1%, and no survival, respectively, after the EV-A71 challenge. Active immunization using CV-A16 provided full protection while WT EV-A71, ChiE71-1-BC and ChiE71-1-EF immunization showed partial cross-protection in CV-A16 lethal challenge with survival rates of 50.0%, 20.0% and 40%, respectively. Disruption of a capsid loop could affect virus immunogenicity, and future vaccine design should include conservation of the enterovirus capsid loops.

## 1. Introduction

Hand, foot and mouth disease (HFMD) is a common disease in children aged below five years old worldwide, especially in the Asia-Pacific region [1]. Enterovirus A71 (EV-A71), coxsackievirus A16 (CV-A16), CV-A2, CV-A4, CV-A6 and CV-A10 are common causative agents of HFMD. Amongst these viruses, EV-A71, CV-A6 and CV-A16 are most frequently associated with HFMD in Malaysia [2,3,4]. Unlike other enteroviruses causing HFMD, EV-A71 can be associated with severe neurological complications, including aseptic meningitis, brainstem encephalitis and pulmonary edema with high fatality rates [5]. To date, no vaccine is available for HFMD-associated enteroviruses other than EV-A71. Furthermore, the inactivated EV-A71 vaccine is currently only marketed in China.

EV-A71 and CV-A16 are positive-stranded RNA viruses under the family of *Picornaviridae* and the genus of *enterovirus*. The genomic RNA comprises a single open reading frame flanked by 5′ and 3′ untranslated regions. Translation of genomic RNA produces a single polypeptide which is further cleaved into structural capsid (VP1-VP4) and non-structural (2A–2C and 3A–3D) proteins [6]. Only VP1-3 capsid proteins are exposed, whereas VP4 is completely buried. Capsids are important for host receptor binding and can trigger host immune responses.

For EV-A71, both IgM and IgG from EV-A71-infected patients recognize all the structural proteins. Of these, the VP1 protein of EV-A71 is the most immunodominant viral protein recognized by HFMD patients [7]. Other than immunodominant binding epitopes, neutralization epitopes are usually located in the capsid surface-exposed loops, where neutralizing antibodies bind, leading to the inactivation of the virus. In other enteroviruses such as the poliovirus, all the neutralizing sites are located at the capsid loops, and recent studies showed that both the D antigen (infectious virus) and C antigen (non-infectious empty particles) are immunogenic [8,9,10]. In total, there are seven well-characterized loops, namely the VP1 BC, DE, EF, GH and HI loops, VP2 EF loop and VP3 GH loop (Figure 1).

Most enterovirus-neutralizing epitopes are located within the capsid loops in the form of conformational or linear epitopes. Both EV-A71 and CV-A16 have some similar neutralizing epitopes at VP1 EF loop (SP55 of EV-A71 and PEP 55 of CV-A16) [7,11,12,13] and GH loop (SP70 of EV-A71 and PEP71 of CV-A16) [12,14]. The PEP27 spanning part of the VP1 DE loop (aa 142–156) was also reported as an IgM-neutralizing epitope of EV-A71 [7]. These previous studies also reported several other CV-A16 neutralizing epitopes within the capsid loops, such as aa 94–108 in the VP1 BC loop and aa 176–190 in the VP3 GH loop. However, these are linear epitopes, and the study on EV-A71 and CV-A16 conformational epitopes is limited and would benefit the future development of broad-spectrum HFMD vaccine design.

Reverse genetics have been used to engineer chimeric EV-A71 (ChiE71) to identify neutralization epitopes, antigenicity and virulence in other picornaviruses [14,15,16]. Previous studies have shown that poliovirus capsid loop engineered with HIV-1 gp41 epitopes could elicit neutralizing antibodies against HIV-1 [17]. Human poliovirus 3 which is one of the three serotypes of poliovirus could also tolerate insertion at the BC loop without affecting virus production [18]. We hypothesized that the capsid loops between enteroviruses can be exchanged and substitution of capsid loops of CV-A16 into EV-A71 will not alter the immunogenicity of EV-A71. In this study, chimeric EV-A71 (ChiE71) carrying capsid loops of CV-A16 were constructed and characterized. We further evaluated the homologous and heterologous immune responses and protection efficacy against ChiE71 in a mouse model. In the end, we will decipher the role of BC and EF loops in the immunogenicity of EV-A71 and CV-A16.

## 2. Materials and Methods

### 2.1. Cells and Plasmids

Rhabdomyosarcoma (RD) cells (ATCC no. CCL136) were grown in Dulbecco’s Modified Eagles Medium (Gibco, Thermo Fisher Scientific, Waltham, MA, USA), supplemented with 10% fetal bovine serum (Thermo Fisher Scientific, Waltham, MA, USA). L-929-hSCARB2 cells (a gift from Satoshi Koike, Tokyo Metropolitan Institute of Medical Science, Tokyo Metropolitan Organization for Medical Research, Tokyo, Japan) were cultured in Dulbecco’s Modified Eagles Medium supplemented with 5% fetal bovine serum and 4 µg/mL of puromycin. Both cells were grown at 37 °C and 5% CO_2_.

### 2.2. Mice

The use of mice in this study was approved by the Institutional Animal Care and Use Committee, Universiti Putra Malaysia (IACUC Number: R046/2020). SPF BALB/c mice were acquired from the Malaysian Institute of Pharmaceuticals and Nutraceuticals and used throughout the study. All animals were housed in a barrier facility in individually ventilated cages at the Comparative Medicine and Technology Unit, Institute of Bioscience, Universiti Putra Malaysia. Mice were provided with sterile water and a standard laboratory rodent diet ad libitum. The mice were monitored daily for health and clinical signs. All procedures performed in the study were in accordance with the guidelines and ethical standards of IACUC.

### 2.3. Construction of Chimera Infections Clones and Virus

The ChiE71 were designed based on the EV-A71 strain 41 (subgenotype B4 virus; GenBank accession number AF316321) and CV-A16 virus strain 22159 (GenBank accession number JQ746673) (Appendix A). The ChiE71 infectious clones were constructed based on previously published pCMV-EV-A71 as the backbone [19]. To substitute the capsid loop sequences on EV-A71 with that of CV-A16, site-directed mutagenesis was performed with pCMV EV-A71 using high-fidelity Q5 polymerase (NEB, Hitchin, UK) and different primer pairs (Appendix A). Corresponding loop regions were substituted, and the blunt end amplified DNA products were ligated into the plasmids. The ligated products were transformed into competent Escherichia coli XL-10 Gold. The plasmids were purified using endotoxin-free PureLink HiPure Plasmid DNA purification mini kit (Invitrogen, Waltham, MA, USA). All constructs were confirmed by Sanger sequencing. The resulting infectious clones were wild types (WT) EV-A71 and WT CV-A16, and clones named after the replaced capsid loop region: pCMV-ChiE71-1-BC, pCMV-ChiE71-1-DE, pCMV-ChiE71-1-EF, pCMV-ChiE71-1-GH, pCMV-ChiE71-2-EF and pCMV-ChiE71-3-GH.

DNA transfection was performed in RD cells using lipofectamine LTX with PLUS reagent (Invitrogen, USA) to generate viable ChiE71. On day 5 post-transfection, the supernatants were harvested by freeze–thawing of transfected RD cells, and reinfection was performed to obtain P1 virus stock.

### 2.4. One-Step Growth Curve and Plaque Morphology

To compare the replication kinetics among ChiE71 and WT viruses, MOI of 0.1 was used to infect RD cells in duplicates. Viral titers of each ChiE71 at 0, 12, 24, 48, 72 and 96 h-post inoculations were determined based on TCID_50_ from the collected supernatants.

A plaque assay was performed to compare the plaque morphologies of WT and ChiE71. Briefly, 1 mL of 10-fold serial diluted ChiE71-1-BC, ChiE71-1-EF, ChiE71-1-GH and WT EV-A71 were added to monolayers of RD cells seeded on a 6-well plate and incubated at 37 °C for 1 h. The inoculum was removed and replaced with 1 mL of 2% FBS DMEM in 0.8% CMC. The plate was incubated for 3 days at 37 °C with 5% CO_2_. The cell supernatant was discarded and rinsed twice with PBS followed by fixation using 3.7% paraformaldehyde at room temperature for 10 min on a rocker. The plate was rinsed followed by staining with crystal violet. The sizes of 10 single plaques of each virus were measured.

### 2.5. Viruses

ChiE71-1-BC, ChiE71-1-EF and WT EV-A71 and WT CV-A16 were concentrated using a centrifugal filter unit (Merck, Darmstadt, Germany). The tubes with virus concentrate and 30% sucrose cushion were loaded to an SW41 Ti Swinging-Bucket Rotor (Beckman Coulter, Brea, CA, USA) and ultracentrifuged at 125,000× *g* for 4 h at 4 °C. The viral pellet was resuspended in 500 µL of PBS. The partially purified viruses were inactivated with 0.25% (*v*/*v*) formaldehyde at 37 °C for 7 days [20,21]. Then, the inactivated viruses were subjected to protein quantification using a bicinchoninic acid (BCA) protein assay kit (Thermo Fisher Scientific, USA).

The lethal dose was first determined for each challenge virus. We used the mouse-adapted EV-A71 strain MP4 (obtained through BEI Resources, NIAID, NIH: enterovirus A71 MP4, NR-472) rather than WT EV-A71, for which we previously reported that the VP1 145Q mutation rendered it non-lethal to mice [15,16]. Survival and clinical symptoms of one-day-old mice were observed following inoculation with MP4 EV-A71 or WT CV-A16 at doses ranging from 10^2^–10^4^ TCID_50_ and 1–0.001 TCID_50_, respectively (Appendix A). The doses for subsequent virus challenge were fixed at 10 LD_50_ of MP4 EV-A71 (10^4^ TCID_50_) and clone-derived WT CV-A16 (0.1 TCID_50_).

### 2.6. Immunization of Adult Mice and Passive Transfer of Anti-ChiE71 Sera

Groups of six-week-old adult female BALB/c mice (n = 5) were immunized with two doses of different inactivated viruses at 14-day intervals. Each dose (100 µL/mouse) consisted of 10 µg of antigen and 3.25 mg of Alhydrogel 2% (InvivoGen, San Diego, CA, USA) as an adjuvant. On the day of immunization, the antigen was mixed with adjuvant and topped up to 100 µL with sterile PBS. The mixture was mixed well with a shaker for 1 h at 4 °C to allow effective adsorption of antigen with adjuvant. The mouse was injected intraperitoneally with the antigen. Booster antigen with the same dose was given to the mice after 14 days. Blood collection was performed on days 0, 14 and 28 through ocular bleeding to obtain serum for further analysis. Anesthesia using 0.1 g/kg of ketamine and 10 mg/kg of xylazine was performed on mice before ocular bleeding.

On day 35, a cardiac puncture was performed to collect the blood from the anesthetized immunized mice. Approximately 1 mL of blood was collected from each mouse. The mice were then euthanized by injection with 1 g/kg of ketamine and 100 mg/kg of xylazine. Mice spleens were collected for cytokine assays.

To evaluate the protective efficacy of passive transfer of antibodies, groups of one-day-old BALB/C mice (n = 5–7) were intraperitoneally inoculated with 100 µL of heat-inactivated pooled serum from the immunized mice. After 1 h, the mice were challenged intraperitoneally with 10 LD_50_ of EV-A71 MP4 or WT CV-A16. The mice were weighed daily and monitored for clinical scores and survival for 14 days. Similar to previous studies [15], the clinical signs were graded as 0, healthy; 1, weak or less active; 2, hunched posture and lethargy; 3, one-limb paralysis; 4, two-limb paralysis; 5, moribund or dead.

### 2.7. Active Immunization in Neonatal Mice

To evaluate the efficacy of ChiE71 in eliciting immune responses in neonatal mice, we performed intraperitoneal immunization of groups of one-day-old mice (n = 5–7) with 1 × 10^5^ TCID50 live ChiE71-1-BC, ChiE71-1-EF or WT EV-A71. Inactivated WT CV-A16 was used for immunization as live CV-A16 is highly lethal to mice. After 7 days, the mice were challenged with 10 LD_50_ of MP4 EV-A71 (10^4^ TCID50) or WT CV-A16 (0.1 TCID50). The mice were monitored for clinical scores and survival for 14 days (Appendix A).

### 2.8. Virus Neutralization Assay

A virus neutralization assay was performed using sera from human healthy donors and adult immunized mice to determine the neutralizing capacity of each serum against WT EV-A71, WT CV -A16 and the ChiE71. In brief, each serum sample was two-fold serial-diluted starting with 1:8 dilution using SFM until 1:256. Volumes of 100 µL sera were mixed with 100 µL of ChiE71 (1 × 10^3^ TCID_50_/mL) and incubated for two hours at 37 °C. The mixture of each dilution was inoculated in triplicates to a monolayer of RD cells seeded on a 96-well plate, followed by incubation for one hour at 37 °C. The inoculum was replaced by 2% FBS DMEM and the cells were monitored for 7 days for the presence of CPE. The neutralizing titer of the serum against the virus was defined as the end-point dilution of serum that inhibited the presence of CPE on cells.

### 2.9. Whole-Virus and Protein VP1 ELISA for IgG Detection

Collected adult mice serum samples were subjected to ELISA to measure total virus-specific IgG. Each well was coated with 10 ng/µL of formaldehyde-inactivated purified WT EV-A71, WT CV-A16 or WT EV-A71 VP1 protein diluted in 50 mM carbonate-bicarbonate buffer (15 mM Na_2_CO_3_, 35 mM NaHCO_3_, pH 9.6). Mouse serum was pooled together and diluted at 1:20 with 1% bovine serum albumin (BSA) in PBS with 0.05% Tween 20 (PBST). After blocking the plate with 5% BSA, diluted serum was incubated for two hours. Horse radish phosphatase (HRP)-conjugated goat anti-mouse IgG (Genetex, Irvine, CA, USA) diluted at 1:10,000 with 1% (*w*/*v*) BSA in PBST was added to the wells and incubated for one hour at 37 °C. After color development with the TMB peroxidase substrate system (KPL, Middletown, DE, USA), the plate was read for absorbance at 450 nm.

### 2.10. Splenocyte Isolation and Th1/Th2 Cytokine Assay

To isolate splenocytes, the spleen was cut into small pieces and gently pressed through a 70 µm strainer. The obtained mouse splenocytes were mixed with 5 mL of RPMI growth medium. The cell suspension was centrifuged at 500× *g* at 4 °C for 5 min. After discarding the supernatant, the cell pellet was resuspended in 5 mL of ammonium-chloride-potassium lysis buffer and incubated on ice for 5 min to lyse the red blood cells. The lysis reaction was stopped by adding 5 mL of cold PBS and the mixture was spun at 500× *g* at 4 °C for 5 min. The pellet was resuspended with 5 mL of growth media. The 96-well cell culture plate was seeded with 3 × 10^6^ cells/well of live splenocytes. To stimulate the splenocytes, 10 µg of inactivated WT EV-A71 or WT CV-A16 was added as an antigen for the splenocyte culture. After culturing for 48 h at 37 °C with 5% CO_2_, the cell supernatants were collected, frozen and stored at −20 °C for the cytokines assay.

The T-helper cell-related cytokines IFN-γ, IL-2, IL-4 and IL-10 produced by immunization in mice were measured using a mouse Th1/Th2 ELISA kit (Invitrogen, USA). After plate coating and blocking, 50 µL of 1:8 diluted splenocyte samples, standards and ELISA diluent as blank was added to the well in replicates and incubated for 2 h. The biotin-conjugated anti-mouse IFN-γ, IL-2, IL-4 and IL-10 antibodies were added followed by incubation for one hour. After washing, HRP-conjugated anti-mouse IgG was added to each well and incubated for 30 min. After color development, the absorbance was read at 450 nm.

### 2.11. Thermal Stability of ChiE71

To determine the thermal stability of ChiE71 particles, we performed a particle stability thermal release assay (PaSTRY) adapted from Walter et al. [22]. Approximately 2 µg of the partially purified virus (with double sucrose gradient ultracentrifugation), including ChiE71-1-BC, ChiE71-1-EF, WT EV-A71 and WT CV-A16, were treated with RNAse followed by RNAse inhibitor to remove residual RNA in the supernatant, leaving only intact virus particles. SYBR green II dye was added as a marker of RNA release. The mixture was analyzed in a StepOnePlus Real-Time PCR System (Thermo Fisher Scientific, USA) and heated from 24 °C to 95 °C. Derivative melt curves of each virus were plotted and RNA release temperatures (Tr) were determined. Tr is defined as the temperature that denotes the minimum negative derivative of fluorescence along the melting curve.

### 2.12. Molecular Modeling

The three-dimensional models of ChiE71 were constructed based on the EV-A71 capsid structure (PDB ID: 3VBS) uploaded to the SWISS-MODEL server [23]. All ChiE71 structure models generated were further characterized and analyzed. ChiE71 capsid structures were first superimposed with that of WT EV-A71 to calculate the average distance between C-α atoms of the chimeric structure model with WT EV-A71 in root mean square deviation (RMSD) using PyMOL version 2.0 (Schrödinger, LLC, New York, NY, USA). To evaluate the flexibility of ChiE71 capsid proteins, the root means square fluctuation (RMSF) of the structures was computed using trajectories collected from 10 ns long molecular dynamics (MD) simulations. All the ChiE71 capsid structures were subjected to protonation state prediction at pH 7.0 using a PDB2PQR server [24] prior to solvation in an octahedral box with TIP3P water models and counterions for charge neutralization. The molecular properties of the structures were described by the ff19SB force field under the AMBER20 program. Trajectory analysis was conducted using the CPPTRAJ module [25].

To predict how the substitution of loops affects the possible binding of ChiE71 with anti-EV-A71 and CV-A16 antibodies, we performed molecular docking simulations to predict the binding structures of ChiE71-1-BC, ChiE71-1-EF, WT EV-A71 and WT CV-A16 with EV-A71 antibody D5 (PDB ID: 3JAU) [26] and CV-A16 antibody NAD97 (PDB ID: 6LHQ) [27] using the HADDOCK server [28,29]. Protein–protein interactions between the antibodies and ChiE71 including salt bridges, hydrogen bonds and electrostatic and hydrophobic charges were dissected and analyzed using the visualization tools UCSF Chimera [30] and Discovery Studio 2020 visualizer version v20.1.0.19295 (BIOVIA, San Diego, CA, USA).

### 2.13. Statistical Analysis

The EV-A71 and CV-A16 specific IgG antibody levels in mice sera were compared using a two-way analysis of variance (ANOVA) with the Bonferroni post hoc test. The Th1/Th2 cytokines levels in mouse serum were compared using one-way ANOVA with Bonferroni post hoc test. Survival rates of mice after the lethal challenge were calculated using the Mantel-Cox log-rank test or Gehan-Breslow-Wilcoxon tests with corrected Bonferroni α value as a post hoc test.

All graphs were plotted, and statistical analyses were completed using GraphPad Prism version 5.0 (GraphPad Software, La Jolla, CA, USA). All quantitative data were presented in mean ± SD. A *p*-value < 0.05 was considered statistically significant.

## 3. Results

### 3.1. Construction of ChiE71 and Structural Modeling

Seven ChiE71 were designed by substituting the EV-A71 capsid loops with the corresponding CV-A16 capsid loops (VP1 BC, DE, EF, GH and HI loop, VP2 EF loop and VP3 GH loop) (Figure 2A). The ChiE71 was named based on the capsid loop region substituted.

Multiple amino acid alignments of capsid loops showed that the VP3 GH loop was the most conserved (91.3%), followed by the VP1 GH loop (87.3%), VP1 EF loop (85.0%) and VP2 EF loop (76.0%). VP1 BC loop was the least conserved with only 45.5% similarity (Figure 1B). The effects of loop exchange on the fluctuation dynamics and flexibility of the capsids were examined. The structural deviation after loop exchange inferred by RMSD of the superimposed protomer structures to 3VBS and corresponding ChiE71 loops were calculated using the software PyMol. Only small structural deviations ranging between 0.064–0.094 Å were observed in all the ChiE71 indicating that the exchanged loops did not affect the protomer structure (Figure 2B). ChiE71-1-BC sequences and RMSF analysis of the loop region are similar to CV-A16 rather than EV-A71, indicating that the exchange of the VP1 BC loop caused the ChiE71-1-BC capsid structure to shift its structural flexibility to higher similarity with CV-A16 (Appendix A). RMSF analyses of ChiE71-1-DE, ChiE71-1-EF, ChiE71-1-GH and ChiE71-1-HI revealed similar flexibility to EV-A71 across the capsids. Conservation is needed for structure stability [31]. Notably, ChiE71-2-EF and ChiE71-3-GH had lower overall RMSF across VP2 and VP3 compared to both EV-A71 and CV-A16, suggesting lower flexibility of these ChiE71.

### 3.2. Growth Characteristics of ChiE71

Other than clone-derived WT EV-A71 and WT CV-A16, we successfully constructed viable ChiE71-1-BC, ChiE71-1-EF, ChiE71-1-GH and ChiE71-3-GH. Three ChiE71, ChiE71-1-DE, ChiE71-1-HI and ChiE71-2-EF; however, they did not yield viable viruses suggesting that the replacement of these loops was lethal. The replication kinetics and plaque morphology of the viable ChiE71 were further analyzed by one-step growth curve and plaque assay, respectively (Figure 3A,B).

The growth curves of all viable ChiE71 showed comparable trends with WT EV-A71 in the first 48 h (Figure 3A). All ChiE71 viruses yielded virus titers of about 10^5^ to 10^7^ TCID_50_/_mL_ on day one post-infection. Over time, the WT EV-A71 yielded higher virus titers of 10^8^ TCID_50_/_mL_ on day three post-infection. The WT EV-A71, ChiE71-1-GH and ChiE71-3-GH produced similar plaque morphologies with a mean plaque size of 2.9 mm (Figure 3B). Compared to WT EV-A71, the ChiE71-1-EF plaque size was 25% smaller while ChiE71-1-BC showed a reduction of up to 80% in plaque size. The altered plaque sizes indicated that the exchange of VP1 EF and BC loop may alter the growth phenotype of EV-A71.

### 3.3. Neutralization Capacity of Human Antisera against ChiE71

Next, we compared the neutralization properties of nine adult human sera against the ChiE71, WT EV-A71 and WT CV-A16 (Figure 3C). The EV-A71 positive sera (serum samples 1–5) cross-neutralized CV-A16 with titers of at least 1:16 suggesting co-presence of anti-CV-A16. Similarly, serum samples 7 and 8, which were negative for anti-EV-A71 antibodies, neutralized CV-A16 at 1:32 and 1:16, respectively. Interestingly, serum sample 8 also could neutralize ChiE71-1-BC and CV-A16 at a titer of 1:32 but not other EV-A71 and other ChiE71. This suggests that the CV-A16 VP1 BC loop has shifted the neutralization properties towards CV-A16. The EV-A71 antisera also neutralized ChiE71-1-BC, ChiE71-1EF and ChiE71-1-GH consistently with similar or 2-8-fold higher titers. The anti-EV-A71-positive human sera demonstrated lower neutralizing capacity against ChiE71-3-GH, suggesting that the exchange of the VP3 GH loop may confer partial neutralization escape. Since the exchange of the VP1 BC loop and EF loop did increase the neutralization capacities of the antisera towards ChiE71, these were chosen for in vivo studies.

### 3.4. Protein–Protein Interactions between ChiE71 and EV-A71 and CV-A16 Antibodies

Prior to the in vivo studies, we determined if the ChiE71 could still bind to EV-A71 D5 and CV-A16 NA9D7 antibodies in silico (Figure 4). D5 and NA9D7 are monoclonal antibodies with well-reported interaction sites and provide 100% protection against lethal EV-A71 and CV-A16 challenge in neonatal mice models, respectively [27,32]. Furthermore, D5 is reported to be able to broadly neutralize EV-A71 [33]. EV-A71 mainly interacted with D5 via VP1 residues 213, 215, 216, 217, 218 and 221 located at the VP1 GH loop, which is conserved and important for hSCARB2 receptor binding (Figure 4C). Similarly, NA9D7 shows broad neutralizing activities against CV-A16 (Figure 4H) [27]. Hence, EV-A71 D5 and CV-A16 NA9D7 were selected as the reference antibodies to study the changes in protein interaction between the chimeric capsid and these antibodies. Compared to the WT EV-A71 which formed 13 hydrogen bonds, ChiE71-1-BC and ChiE71-1-EF formed 7 and 11 hydrogen bonds with antibody D5, respectively. Within the WT EV-A71-D5 docked complex, residue 218 has the most interactions with D5 through hydrogen bonds, salt bridge and Van der Waals interaction (Figure 4C). EV-A71 and both ChiE71-1-BC and ChiE71-1-EF formed common interaction sites when docked with D5, including residues 215, 216, 217, and 220 which are all located within the VP1 GH loop. The total interactions and lesser hydrogen bonds on ChiE71-1-BC-D5 complex compared to WT complex indicate a weaker binding with D5 (Figure 4A,B). Similar to ChiE71-1-BC, ChiE71-1-EF consists of lesser interaction sites and loss of K218 as an important interaction site for the D5 antibody (Figure 4B). However, the interaction of ChiE71-1-EF could be better than the CHiE71-1-BC due to more hydrogen bonds. We also simulated the CV-A16 virus capsid complexing with antibody D5, showing 2 hydrogen bonds and 5 Van Der Waals interactions, with no strong intermolecular interactions such as salt bridges or attractive charges, indicating a lack of binding affinity towards D5 (Figure 4D). Taken together, ChiE71 still binds to the D5 antibody, but with altered epitope–antibody interactions.

On the other hand, NA9D7 mainly interacts with CV-A16 at the VP1 GH loop as well. However, unlike D5 with residue 218 as the main interaction site when bound to WT EV-A71, the main interaction site of NA9D7 and CV-A16 are 216 and 220 with hydrogen bond and alkyl interactions (Figure 4H). The common interaction sites of ChiE71-1-BC- and ChiE71-1-EF-NAD97 complexes were VP1 residues 220 (Figure 4E,F). The ChiE71-1-BC-NA9D7 complex consists of two hydrogen bonds which was similar to the CV-A16-NA9D7 complex (Figure 4E,H). CHiE71-1-EF-NA9D7 interacts via VP1 residue 220, 221, and VP2 residue 134, 136. ChiE71-1-EF is bound with mainly hydrogen bond and alkyl interactions (Figure 4F). CV-A16 mainly interacted with NA9D7 via VP1 residues 216, 217, 220 and 224 (Figure 4H). Notably, WT EV-A71 can bind to NA9D7 with stronger interactions than the CV-A16-NA9D7 complex due to hydrogen bonds and salt bridge interactions from charged amino acid E217 (Figure 4G). Taken together, NA9D7 interacts with EV-A71 through different epitopes within the VP1 GH loop.

### 3.5. Thermal Stability of ChiE71

The partially purified ChiE71 were analyzed by Western blot with monoclonal antibody MAb979 (Figure 5A). VP0 (36 kDa) and VP2 (28 kDa) proteins were detected in WT EV-A71, ChiE71-1-BC and ChiE71-1-EF but not in WT CV-A16 (Figure 5A). Notably, the VP2 protein was less abundant in ChiE71-1-BC and ChiE71-1-EF. This indicates that ChiE71-1-BC with CV-A16 VP1 BC loop and ChiE71-1-EF with CV-A16 EF loop may have affected the maturation of the viral proteins.

To evaluate the thermal stability of the selected ChiE71, we performed PaSTRY with the double sucrose gradient purified viruses to determine the RNA release of the viral genome over a temperature gradient. As the temperature increases from 24 to 95 °C, viral RNA will be released due to the unfolding of virus capsid protein. A high Tr indicates higher thermal stability of capsid structure, as a higher temperature is needed for the unfolding of capsid and release of genomic RNA. ChiE71-1-BC has Tr = 61.14 °C, WT EV-A71 Tr = 59.75 °C, CV-A16 Tr = 59.75 °C and ChiE71-1-EF Tr = 54.19 °C (Figure 5B). This suggests that ChiE71-1-BC has better thermal stability while ChiE71-1-IEF is lower. The increase in ChiE71-1-BC capsid thermal stability could reduce RNA uncoating and may explain the reduction in plaque size of ChiE71-1-BC (Figure 3).

### 3.6. ChiE71 Induces Humoral Immune Responses against Both EV-A71 and CV-A16

Groups of five female mice (6–8 weeks old) were subjected to 10 µg/dose of immunization of the inactivated viruses ChiE71-1-BC, ChiE71-1-EF, WT EV-A71 and WT CV-A16, and PBS as control (Figure 6A). EV-A71-specific and CV-A16-specific IgG levels of mice sera were evaluated using ELISA. As expected, PBS-immunized mice did not produce any virus-specific IgG. Both anti-EV-A71-specific and CV-A16-specific antibodies were detected in mice sera after the first immunization with all viruses and antibody levels remaining high after the booster immunization (Figure 6B).

The immunized mouse sera collected through cardiac puncture were pooled in equal parts for neutralization assay against EV-A71 and CV-A16. Sera from all the viruses including both ChiE71-1-BC and ChiE71-1-EF neutralized CV-A16 at a titer of at least 1:16 (Figure 6C).

ChiE71-1-BC serum has a partial loss of the neutralizing capacity against EV-A71 but gained neutralizing capacity against CV-A16. Compared to the EV-A71-immunized mouse serum, the ChiE71-1-BC-immunized mouse pooled serum had a two-fold reduction of neutralizing capacity against WT EV-A71 (Figure 6C). This result indicates that the antibody elicited by ChiE71-1-BC has lowered neutralizing activity towards EV-A71, similar to the earlier reduction of neutralizing capacity shown against the human sera. Interestingly, compared to WT EV-A71, ChiE71-1-EF-immunised pooled serum obtained higher cross-neutralizing titers of 1:16 against CV-A16, which is comparable with CV-A16-immunised mouse serum, while neutralizing EV-A71 at the same titer of 1:32 (Figure 6C). This suggests that the chimera virus with the exchange of VP1 EF loop did not have compromised neutralizing capacity against EV-A71, in addition to the gain of cross-neutralization against CV-A16.

The EV-A71 VP1-specific IgG levels were detected by performing ELISA with coated EV-A71 VP1 protein (Figure 6D), and ChiE71-1-EF and EV-A71 immunized mice serum contained a high level of EV-A71 VP1-binding IgG. This indicates ChiE71-1-EF elicited IgG that can cross-react to EV-A71 VP1. ChiE71-1-BC and CV-A16 immunized mouse serum contained very low levels of EV-A71 VP1 binding IgG, indicating the IgG in these two sera did not bind to EV-A71 VP1 protein.

### 3.7. ChiE71 Induce Cellular Immune Responses against EV-A71 and CV-A16

To evaluate the cellular immune responses induced by ChiE71 in adult mice, the concentrations of the Th1 type (IFN-γ and IL-2) and Th2 type (IL-4 and IL-10) cytokines in immunized mice spleens following viral stimulation were further analyzed using ELISA. Compared to the control group, all mice immunized with inactivated EV-A71, CV-A16, ChiE71-1-BC and ChiE71-1-EF elicited high IFN-γ, IL-2, IL-4 and IL-10 production (Figure 6E). Comparing ChiE71-1-BC, ChiE71-1-EF, CV-A16 and EV-A71 immunized mice, the IL-2 level detected in ChiE71-1-BC was the highest, although not significant. After stimulation by either EV-A71 or CV-A16 antigen, IL-4 levels detected in ChiE71-1-EF were the lowest among the four groups, though not significant. There was no significant difference between IL-10 levels induced by ChiE71-1-BC, ChiE71-1-EF, EV-A71 and CV-A16 immunization. Notably, the IFN-γ level against CV-A16 in ChiE71-1-EF immunized mouse spleens was the lowest and significantly lower than that of ChiE71-1-BC immunized mouse spleens. Compared to PBS, WT EV-A71 and CV-A16 had significantly higher Th1 and Th2 responses against respective corresponding antigens. Taken together, active immunization using ChiE71-1-BC, ChiE71-1-EF, WT EV-A71 and WT CV-A16 elicited good Th1 and Th2 responses in mice.

### 3.8. Passive Transfer of Chi-E71-Immunized Mouse Sera

To evaluate the protective efficacy of the sera derived from mice immunized with ChiE71, groups of one-day-old neonatal mice were passively transferred with antisera and challenged 1 h later with either WT CV-A16 or MP4 EV-A71. The mice were then monitored for clinical symptoms for 14 days (Figure 7A). The control group of mice injected with PBS-immunized mouse sera showed severe symptoms and 100% mortality upon MP4 EV-A71 or WT CV-A16 challenge. After the MP4 EV-A71 challenge, the neonatal mice injected with WT EV-A71-immunized mouse sera exhibited no clinical symptoms, all the mice survived and were protected from disease. However, the neonatal mice injected with sera from the ChiE71-1-EF- and ChiE71-1-BC- immunized mice showed 80% and no survival, respectively, after the MP4 EV-A71 challenge (Figure 7B). This indicates that the substituted CV-A16 loops especially the BC loop altered the EV-A71 immunogenicity. When the neonatal mice were challenged with CV-A16, the survival rate of injected with CV-A16-immunized mouse sera was 40% at 14 days post-infection, suggesting mice only retained partial protection against the lethal CV-A16 challenge (Figure 7B). Similarly, the neonatal mice receiving sera from ChiE71-1-BC-immunized had 42.9% survival after the CV-A16 challenge, but this was only 20.0% in ChiE71-1-EF (Figure 7B). Due to small sample sizes, the difference was not statistically significant.

### 3.9. Active Immunization of ChiE71 in Neonatal Mice

For active immunization, one-day-old neonatal mice were immunized with 10^5^ TCID_50_ of WT EV-A71, inactivated CV-A16, ChiE71-1-BC or ChiE71-1-EF (Figure 7C). Better protection was observed with active immunization than with passive immunization. Upon MP4 EV-A71 challenge, ChiE71-1-BC, ChiE71-1-EF and EV-A71-immunized mice achieved 60.0%, 57.1% and 62.5% of survival, respectively (Figure 7D). This indicates that both ChiE71 retained partial immunogenicity upon MP4 EV-A71 lethal challenge, similar to the WT EV-A71. In contrast, no CV-A16 immunized neonatal mice survived the MP4 EV-A71 challenge (Figure 7D).

As expected, all CV-A16-immunized neonatal mice survived the CV-A16 lethal challenge. Interestingly, EV-A71 immunization cross-protected neonatal mice from the CV-A16 challenge with a survival rate of 50% suggesting natural cross-protection (Figure 7D). Both ChiE71-1-BC and ChiE71-1-EF-immunized neonatal mice were partially protected from CV-A16 lethal challenge with survival rates of 20% and 40%, respectively (Figure 7D). This suggests that the substitution of CV-A16 loops especially the BC loop reduced the natural cross-protection of EV-A71 against CV-A16.

## 4. Discussion

Both EV-A71 and CV-A16 shared >85% nucleotide and 90% amino acid similarities suggesting the possibility of substituting their genes without lethal effects. In this study, the capsid loops of EV-A71 were substituted with the corresponding loops of CV-A16. Our in vivo study in mice further demonstrated the importance of VP1 BC and EF loops as epitopes of EV-A71 and CV-A16. Seven ChiE71 were constructed, of which four were viable (ChiE71-ChiE71-1-BC, ChiE71-1-EF, ChiE71-1-GH and ChiE71-3-GH), confirming that EV-A71 VP1 BC, EF, GH and VP3 GH capsid loops could be replaced with closely related species of enterovirus such as CV-A16. 

Sequence conservation, fluctuation dynamics and capsid thermostability could affect the viral infectivity and virulence [31,34,35,36]. Interestingly, the viability of the chimeras did not correlate with sequence conservation as ChiE71-BC with the highest differences was still viable but not the more conserved ChiE71-2-EF. The four ChiE71 were viable, the ChiE71 had changes in plaque growth, structural stability and flexibility, thermal stability and immunogenicity. We monitored the sequence adaptations over five passages, and all ChiE71 had no sequence changes and were stable (data not shown). Capsid loops are often exposed on the virus particle surface and play multiple roles including cell tropism, receptor binding and virus uncoating and act as antigenic sites to stimulate immune responses [37]. Specifically, VP1 BC and HI loops are associated with neural cell tropism [38,39]. In poliovirus, these loops also control host range and mouse neurovirulence [40,41,42]. We did not test these roles specifically in this study, but when the BC loop of EV-A71 was substituted with the corresponding region of CV-A16, it shifted the flexibility of loop structure and thermal stability and affected antibody neutralization and immune protection. Consistently, ChiE71-1-BC-immunized mouse sera provided no protection against EV-A71 challenge despite a high amount of IgG and good cellular Th1 and Th2 responses but had partial protection against CV-A16. In contrast, active immunization of ChiE71-1-BC, which enabled stimulation of both humoral and cellular immune responses was able to protect against EV-A71 but with much lower protection against CV-A16 challenge. These results suggest that the BC loop is important for the immunogenicity of EV-A71 and CV-A16 and provide virus-specific immune protection against EV-A71 and CV-A16, respectively. Here, we also showed that both passive and active immunization of ChiE71-1-EF with the CV-A16 VP1 EF loop provided similar immune protection against EV-A71, behaving like the WT virus. The EF loop maintains the cross-protective immune responses against EV-A71 and CV-A16. As the VP1 EF loop is highly conserved in EV-A71 and CV-A16 (85.0% identical) and with minimal changes in structure flexibility, the antibodies could neutralize against both EV-A71 and CV-A16 challenges. This is expected as previous studies reported that the VP1 EF loop contains conserved epitopes of both EV-A71 and CV-A16 [11,43].

Both EV-A71 and CV-A16 cause HFMD, and our data showed that active immunization of WT EV-A71 also protected against the CV-A16 challenge. This natural cross-protection has also been previously reported [4,44,45,46]. However, real-life effectiveness study of the monovalent EV-A71 vaccine in China showed reduced severe HFMD cases but found more cases of CV-A16 and other enteroviruses, suggesting limited or short-lived cross-protection between these enteroviruses [47]. Hence, the development of multivalent vaccines which cover broad enterovirus serotypes will be needed to further reduce the burden of other enterovirus-associated HFMD.

Findings from this study showed that capsid loops are important epitopes, and the information can be used to develop future HFMD vaccines. Both EV-A71 and CV-A16 in the form of inactivated viruses or virus-like particles inoculated in monkeys and mice provided dual humoral immune responses [48,49]. These previously co-administered, bivalent, trivalent or tetravalent HFMD vaccines showed strong neutralization antibody responses and no immune interference between the different enteroviruses [49,50,51]. Chimeric flavivirus multivalent vaccine candidates with different backbones such as dengue 2 virus, Japanese encephalitis virus and yellow fever virus, and genes encoding envelope proteins replaced with corresponding genes of another flavivirus have been successfully developed [52,53,54]. ChiE71-1-BC and ChiE71-1-EF stimulated good Th1 and Th2 responses (similar to the WT), but substitution with CV-A16 loops in EV-A71 altered the immunogenicity of EV-A71 and reduced the natural cross-protection against CV-A16. As capsid loops are important for immunogenicity, future vaccine design should be carefully designed for the conservation of the enterovirus capsid loops.

A limitation of the study is that full protection could not be achieved by WT EV-A71 active immunization against the MP4 EV-A71 challenge. This could be due to the use of mouse-adapted MP4 belonging to genotype C4 while WT EV-A71 belongs to genotype B4. This study only explored both capsid BC and EF loops. Another potential loop is the VP1 GH loop, a SCARB2 receptor binding site, that acts as a sensor-adaptor during uncoating and is highly conserved [55]. Both EV-A71 and CV-A16 also utilize SCARB2 as an entry receptor [56] and therefore the substitution is unlikely to affect the receptor binding.

A study reported that the replacement of multiple CV-A16 epitopes into EV-A71 virus-like particles elicited higher neutralizing antibody titer than the use of a single epitope [57]. Other studies have reported that many EV-A71 and CV-A16 capsid loops act as epitopes including VP1 EF, GH and VP2 EF loops [11,51,52,58,59]. Further investigations should include the simultaneous addition of capsid loops on EV-A71 or CV-A16 to provide additional cross-protection. ChiE71 can also be used for future chimera applications, for example, the engineered poliovirus chimera with HIV epitopes elicited good HIV-1 neutralizing antibodies [17] and the polio-rhinovirus chimera (PVSRIPO) as oncolytic virus therapy for glioblastoma [60,61].

In conclusion, this is a proof-of-concept study demonstrating the feasibility of replacing corresponding capsid loops between enteroviruses to study the role of these loops in protective immune responses involving both humoral and cellular. Both VP1 BC and EF loops contribute to the immunogenicity of EV-A71 and the natural cross-protection of EV-A71 against CV-A16.

## Figures and Tables

**Figure 1 vaccines-11-01363-f001:**
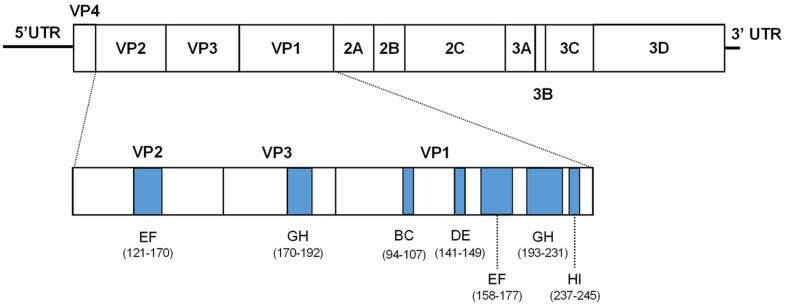
Schematic representation of EV-A71 genome and architecture of capsid loops. Structural protein consists of VP1–VP4. Blue-colored regions represent the location of each capsid loop constructed in this study, and the numbers indicate the amino acid position within each viral protein.

**Figure 2 vaccines-11-01363-f002:**
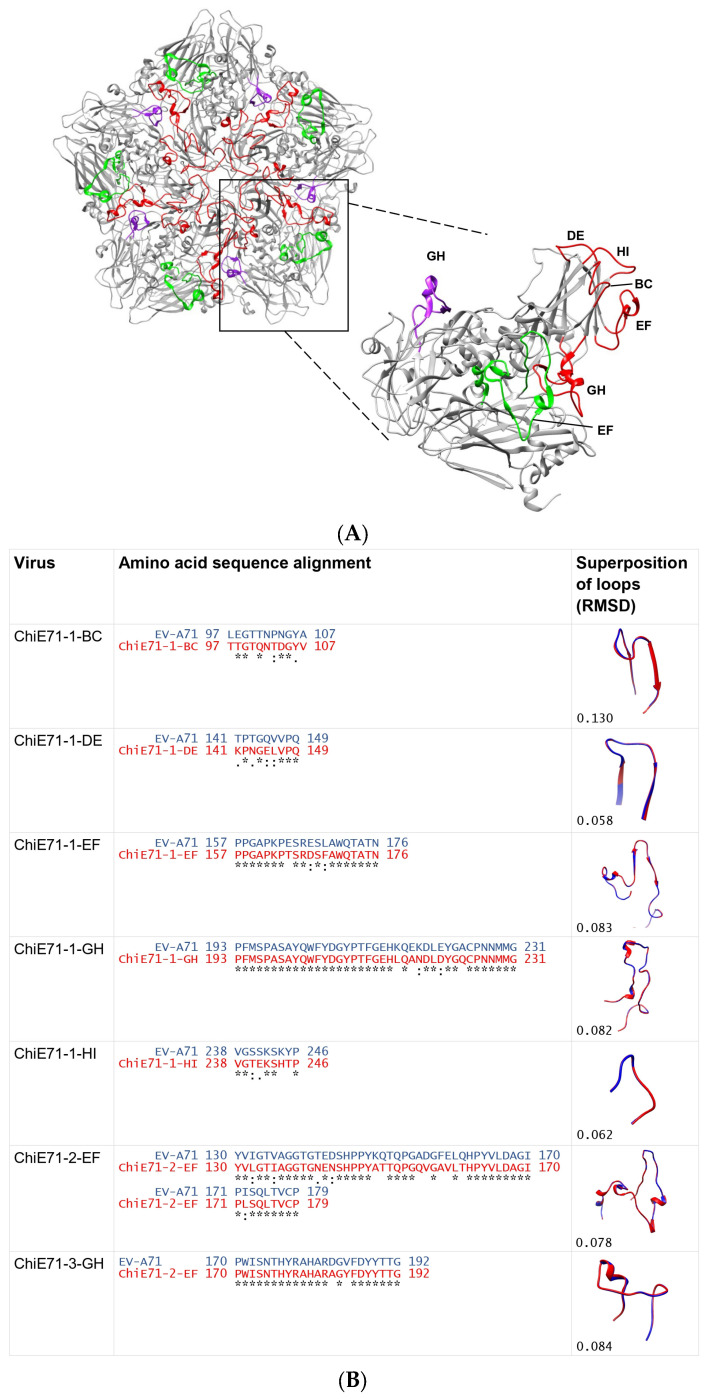
(**A**). Schematic representation of location of capsid loops involved in this study. The figure illustrates pentamer and protomer model of EV-A71 (PDB ID: 3VBS) with highlighted capsid loops involved in this study, including VP1 BC, DE, EF, GH and HI loops (red), VP2 EF loop (green) and VP3 GH loop (purple). (**B**). Loop sequences of ChiE71 substituted by CV-A16. Columns from left to right represent the multiple sequence alignments and stereo view superpositions of corresponding loop regions of VP1 of WT EV-A71 (blue) and ChiE71 (red). The ChiE71 designed in this study include ChiE71-1-BC, ChiE71-1-DE, ChiE71-1-EF, ChiE71-1-GH, ChiE71-1-HI, ChiE71-2-EF and ChiE71-3-GH. The alignments were performed using Clustal Omega. ((∗)—identical conserved residue; (:)— strongly similar residues; (.)—residues have weakly similar properties; none—residues that are not conserved). The superpositions of ChiE71-1-BC, ChiE71-1-DE, ChiE71-1-EF, ChiE71-1-GH, ChiE71-1-HI, ChiE71-2-EF and ChiE71-3-GH (red) with WT EV-A71 (blue) showing structural variations upon replacement with the corresponding CV-A16 VP1 BC, DE, EF, GH, HI, VP2 EF and VP3 GH loops, respectively.

**Figure 3 vaccines-11-01363-f003:**
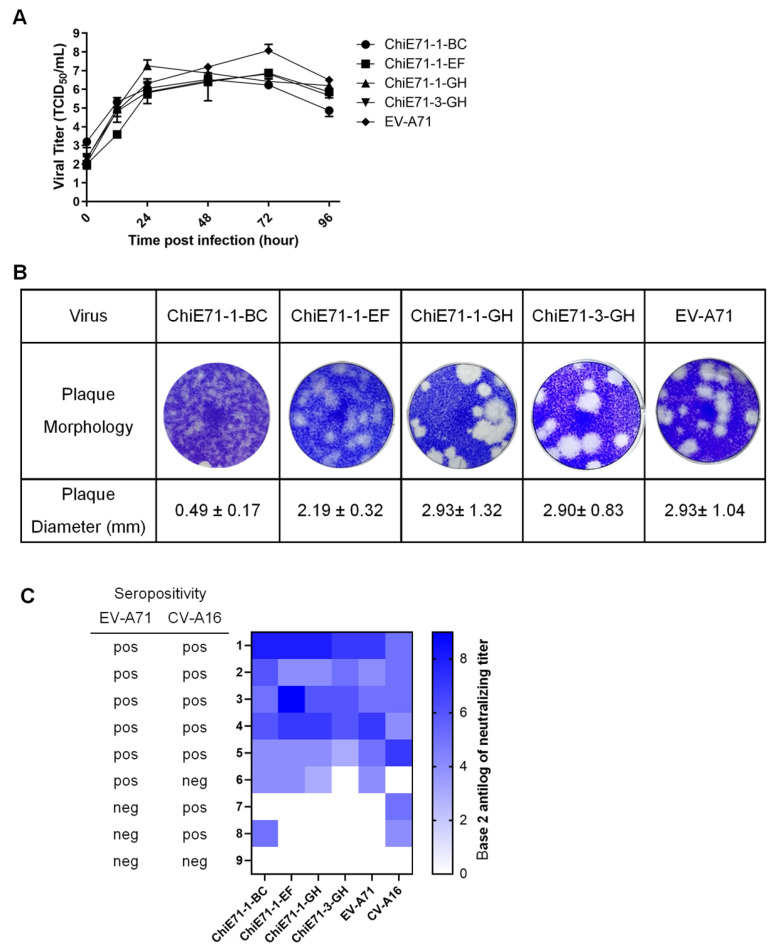
Characteristics of ChiE71. (**A**) One-step growth curves of the ChiE71. The RD cells were infected with 0.1 MOI of WT EV-A71 and ChiE71. The cell supernatants were harvested at 0, 12, 24, 48 and 96 h post-infection and viral titers were determined. The data are presented as mean of three replicates with standard deviation as the error bar. (**B**) Plaque morphology and plaque diameters of the WT EV-A71 and ChiE71 at passage 1 in RD cells. Data are presented as a mean of at least five replicates with standard deviation. (**C**) Heat map of neutralization capacity of each human anti-EV-A71 seropositive or seronegative serum against ChiE71, EV-A71 and CV-A16. The nine serum samples are shown on the left vertical axis, samples 1–6 are anti-EV-A71 positive and 7–9 are seronegative. The color gradient represents anti-log of base 2 neutralization titer. The lowest serum dilution was 1:2^3^.

**Figure 4 vaccines-11-01363-f004:**
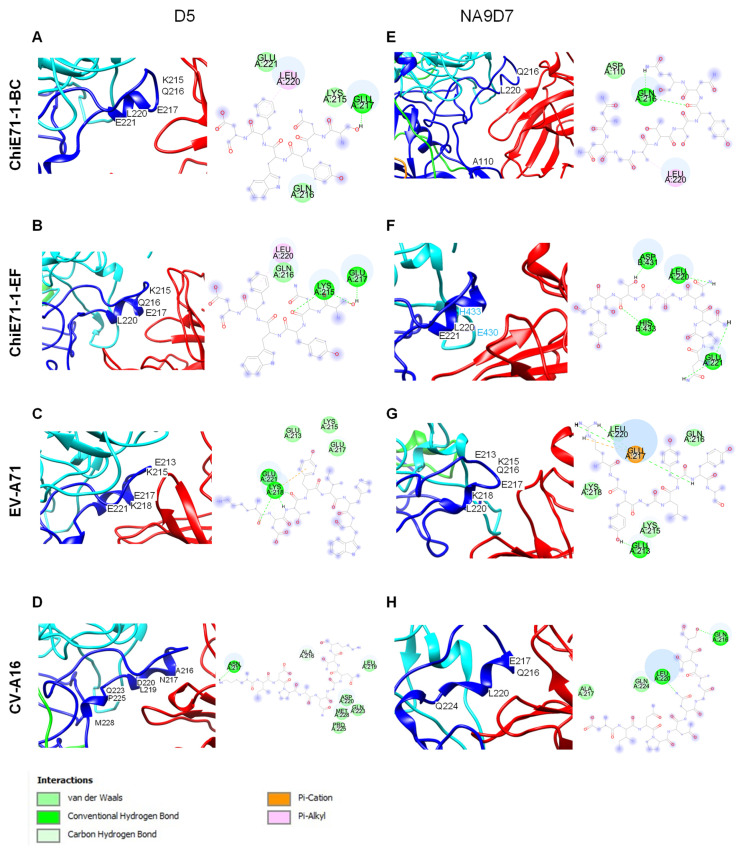
Type of interactions and sites of docked chimeras with anti-EV-A71 or anti-CV-A16 mono-clonal antibodies. WT EV-A71, WT CV-A16 and ChiE71-1-BC and ChiE71-1-EF models were docked with anti-EV-A71 D5 (PDB: 3JAU), in (**A**–**D**); or anti-CV-A16 D110 (PDB: 6LHQ), in (**E**–**H**). Each ChiE71 is presented as a protomer with four capsid proteins VP1 (blue), VP2 (cyan), VP3 (green) and VP4 (orange). Docked antibodies are presented as red strands.

**Figure 5 vaccines-11-01363-f005:**
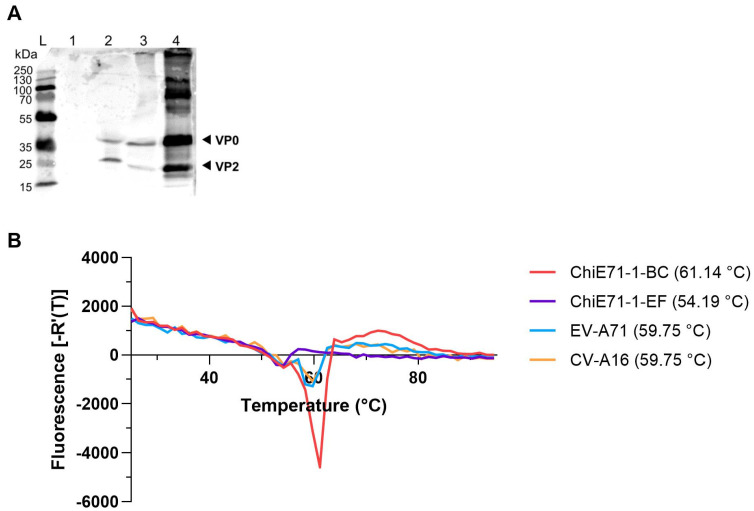
Characterization of ChiE71-1-BC and ChiE71-1-EF. (**A**) Western blot of partially purified WT CV-A16 (lane 1), WT EV-A71 (lane 2), ChiE71-1-BC (lane 3) and ChiE71-1-EF (lane 4). (**B**) Particle stability thermal release assay (PaSTRY). Partially purified viruses were mixed with SYBR green II and heated from 25 to 95 °C in a StepOne Plus Real-Time PCR System. Derivative melt curves for ChiE71-1-BC, ChiE71-1-EF, EV-A71 and CV-A16 were plotted.

**Figure 6 vaccines-11-01363-f006:**
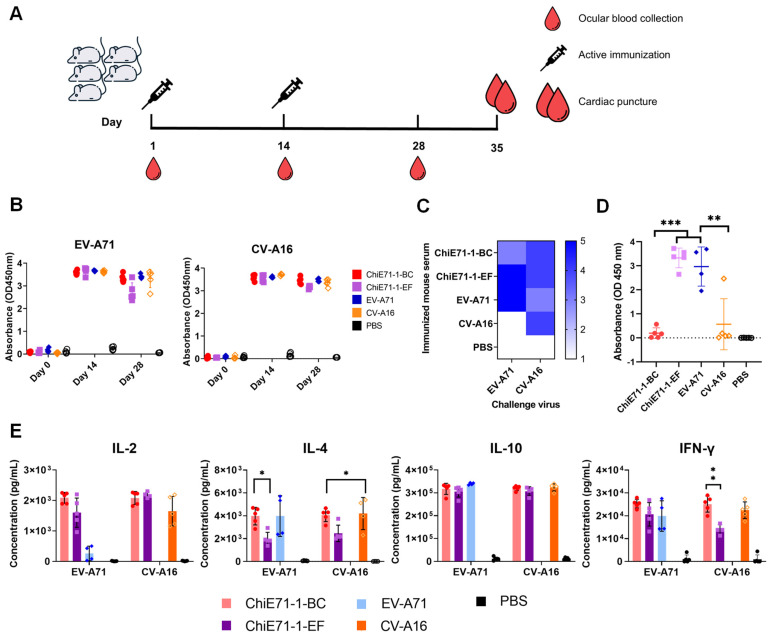
Immunogenicity of inactivated viruses in adult mice. (**A**) Schematic of the in vivo experiment to evaluate immune responses following ChiE71-1-BC and ChiE71-1-EF immunization. Each group of female adult BALB/C mice (n = 5) received prime injection and one booster of 10 μg of inactivated virus at indicated times. (**B**) Virus-specific IgG analysis of immunized mice sera on days 0, 14 and 28 using ELISA. (**C**) Heat map of EV-A71 and CV-A16 neutralization titers of immunized mice sera (after booster immunization). The lowest dilution of serum was 1:2^3^. (**D**) EV-A71 VP1-specific IgG analysis of immunized mouse serum. The day 28 immunized mice sera were subjected to EV-A71 VP1-coated ELISA. (**E**) Inactivated ChiE71, WT EV-A71 and WT CV-A16 cellular immune responses in mice. Immunized mouse splenocytes were stimulated by 1μg of inactivated WT EV-A71 or WT CV-A16. The cytokine (IFN-γ, IL-2, IL-4 and IL-10) concentrations in the cell culture supernatant were detected using ELISA. Each dot represents serum from an individual mouse. Data are presented as mean (±SD) of mice in each group. *** *p*<0.001, ** *p* < 0.01; * *p* < 0.05.

**Figure 7 vaccines-11-01363-f007:**
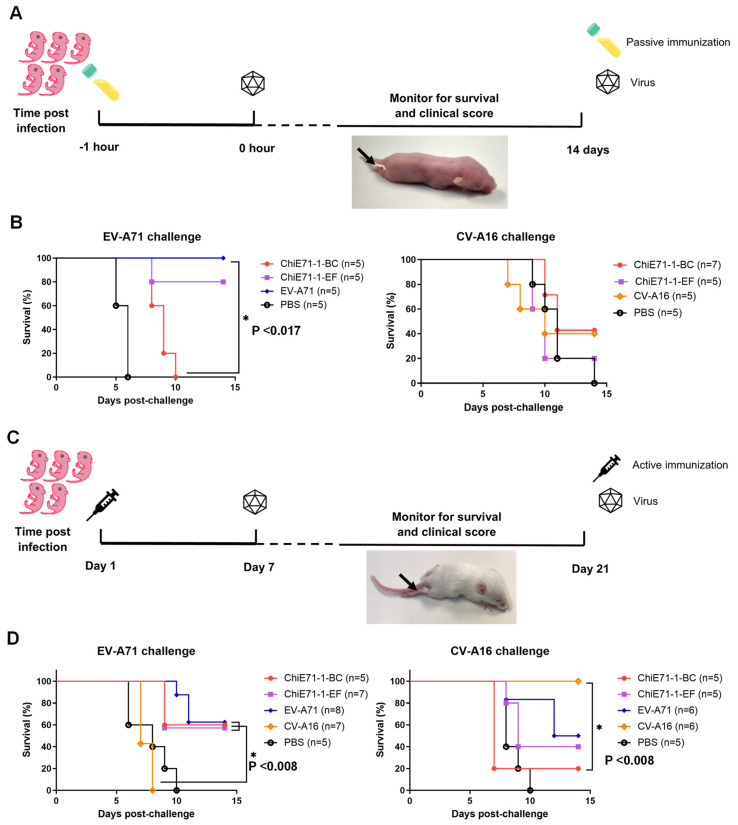
In vivo protection following ChiE71 immunization of neonatal mice. (**A**) Schematic of in vivo experiment involving passive transfer of one-day-old neonatal mice with ChiE71-1-BC-, ChiE71-1-EF-, WT EV-A71- or WT CV-A16-immunized mouse sera followed by lethal MP4 EV-A71 or CV-A16 challenge. (**B**) Survival curves of the mice after passive transfer (**C**) Schematic of in vivo experiment involving active immunization of one-day-old neonatal mice with live ChiE71-1-BC, ChiE71-1-EF, WT EV-A71 or inactivated CV-A16 followed by lethal MP4 EV-A71 or CV-A16 challenge on day 7. (**D**) Survival curves of the mice after active immunization. Survival curves were compared, and the significance was determined based on *p*-values of Gehan-Breslow-Wilcoxon tests with corrected Bonferroni α values. Asterisk (*) indicates a significant difference.

## Data Availability

All data is contained within the article and Appendix A.

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
