# Peer review of "Substitution of Coxsackievirus A16 VP1 BC and EF Loop Altered the Protective Immune Responses in Chimera Enterovirus A71"

_vaccines, 2023, doi:10.3390/vaccines11081363_

Round 1

Reviewer 1 Report

The review article under title “Substitution of coxsackievirus A16 VP1 BC and EF loop altered 2 the protective immune responses in chimera enterovirus A71” by Tan and coworkers presents results on the substitution of coxsackieviruses in the active immune response in chimera enterovierus. The article could be of potential interest to the readers of the Vaccines My recommendation is MAJOR REVISION as there are several points that should be addressed by the authors, as shown below.

The authors should answer the following:

1.     The Authors should check the length of the Abstract as there is a limitation to 200 words

2.     What could be the reason for the loop conservation and how is this affected by the structure in section 3.1?

3.     What type of interactions were present between antibodies and protein structure in section 3.4?

4.     How these interactions depend on structure of antibodies?

5.     What induced the difference in Tm values for the studied compounds?

6.     The authors should compare the survival rates to other available methods.

Author Response

1. The Authors should check the length of the Abstract as there is a limitation to 200 words

We have cut down to 212 words. We hope this is acceptable now.

 2. What could be the reason for the loop conservation and how is this affected by the structure in section 3.1?

The conservation of the loops is influenced by receptor binding, immune evasion, host range and structure stability. VP1 surface loops are receptor binding sites for most picornaviruses. The capsid loops surrounding the canyon (binding pockets) include VP1 BC, EF, DE, GH and HI loops. Capsid loops also contribute to conformational epitopes and antibody neutralization.

Pararagraph 2-4 in introduction have addressed this.

Capsid loops have high fluctuation dynamics (high RMSF). Lower binding energy and lower flexibility will favour stronger interactions between the virus and antibody complex. We have shown the RMSF in Supplementary Figure S3.

A recent paper has showed that to maintain structure stability during evolution, proline and arginine of VP1 protein of enteroviruses are conserved. We have added “The conservation is needed for structure stability” into results 3.1 (line 338).

3. What type of interactions were present between antibodies and protein structure in section 3.4?

We have added the type of interactions between the antibodies and ChiE71, WT EV-A71 and CV-A16. This is now in revised Figure 4 (showing interactions) and section3.4 write up.

4. How these interactions depend on structure of antibodies?

Different ChiE71 bind to the D5 and NAD97 with different type of interactions. This is now in revised Figure 4 and section3.4.

5. What induced the difference in Tm values for the studied compounds?

The term Tm were replaced with a more suitable term "Tr" to indicate temperature at which RNA release occurs and bind to SYBR green II dye.  As the temperature increases, the virus capsid loss its conformation, unfold, expose the hydrophobic regions concealed in the native capsid and release its genome. This assay is used to characterize virus capsid stability and genome release for development of thermostable vaccine.

Many studies have showed that thermostable virus (high Tr) is usually attenuated , making it a good vaccine candidate. See papers:

  1. Nguyen et al., 2019. Identification and Characterization of a Poliovirus Capsid Mutant with Enhanced Thermal Stability. doi: 10.1128/JVI.01510-18
  2. Fox et al., 2017. Genetically Thermo-Stabilised, Immunogenic Poliovirus Empty Capsids; a Strategy for Non-replicating Vaccines. doi: 10.1371/journal.ppat.1006117

Many factors could change the Tr and affect infectivity and immunogenicity as showed in previous study:

  1. Increased particle thermotolerance could decreased efficiency of cell entry but increased virulence in mice (https://www.biorxiv.org/content/10.1101/2022.09.28.506941v1.full)
  2. EV-A71 infectious clone carrying conserved VP1-P157A replicated well but showed smaller plaque size, lower viral growth kinetics, and thermal instability at 39.5°C when compared to the WT. (https://www.mdpi.com/1999-4915/14/2/364)
  3. Emergence of thermally resistant EV-A71 virus with repeated passage and incubation at 52.5 °C for 30 min. (https://www.microbiologyresearch.org/content/journal/jgv/10.1099/jgv.0.001771)

6. The authors should compare the survival rates to other available methods

We have added more data, 1. the clinical scores and 2. body weight of the mice. Please see Supplementary Fig 4.

Reviewer 2 Report

The authors explored the immunological importance of capsid loops in EV-A71 and CV-A16 by constructing chimeric EV-A71 (ChiE71) carrying capsid loops of CV-A16 and evaluating the immune responses induced by ChiE71, as well as their protection efficacy against EV71 an CV-A16 in a mouse model. Although I have to notice that substitution with CV-A16 VP1 BC loop was insufficient to confer protection against CV-A16, their study is helpful to the deep understanding of the roles of EV capsid proteins in the protective immune responses against the viruses and also to the development of vaccines for newly discovered enteroviruses.

Here are some questions and suggestions 

1. ... and substitution of capsid loops of CV-A16 into EV-A71 will not alter its immunogenicity. (line 80) This sentence is not clear. Does the authors mean the substitution would confer the immunological characteristics of CV-A16 to the ChiEV71 ? 

2. The results of PaSTRY (line 411-417, Fig 4.B) needs more explanation for the readers to understand. 

3. Figure 2. C : It is better to label the information of samples on the figure directly (samples 1-6 are anti-EV-A71 positive and 7-9 are seronegative). The response of Sample 8 to ChiE71-1-BC needs more explanation.

4. How did the authors explain that ChiE71-1-BC, ChiE71-1- EF, WT EV-A71 and WT CV-A16 elicited good Th1 and Th2 responses in mice (line 484-485).  The authors should show some reported Th epitopes in the capsid loop and the difference in the epitopes between EV71 and CV-A16 

5. The authors need to explain the significance of the test on thermal stability of ChiE71. What is the relation of this assay with the immune responses induced by the chimeric EVs? 

6. The legends of Fig. 6 is too simple to clearly describe the results of the passive immunization (line 528). The authors should add challenge to the subtitle of Fig. 6B as 6D.

Author Response

1. “... and substitution of capsid loops of CV-A16 into EV-A71 will not alter its immunogenicity.” (line 80) This sentence is not clear. Does the authors mean the substitution would confer the immunological characteristics of CV-A16 to the ChiEV71 ?

The sentence was rephrased. We are interested to study if the loop exchange will alter EV-A71 immunogenicity. See line 97.

2. The results of PaSTRY (line 411-417, Fig 4.B) needs more explanation for the readers to understand.

We have added a few lines and appropriate citation (See line 267-275).

3. Figure 2C : It is better to label the information of samples on the figure directly (samples 1-6 are anti-EV-A71 positive and 7-9 are seronegative). The response of Sample 8 to ChiE71-1-BC needs more explanation.

Thank you. We have relabelled the figures.

Further explanation for serum sample 8 was added “Interestingly, serum sample 8 also could neutralize ChiE71-1-BC and CV-A16 at a titer of 1: 32 but no other EV-A71 and other ChiE71. This suggests that the CV-A16 VP1 BC loop have shifted the neutralization properties towards CV-A16 “ (line 400-402.

4. How did the authors explain that ChiE71-1-BC, ChiE71-1- EF, WT EV-A71 and WT CV-A16 elicited good Th1 and Th2 responses in mice (line 484-485).  The authors should show some reported Th epitopes in the capsid loop and the difference in the epitopes between EV71 and CV-A16

From the Figure 6E, PBS-inoculated mice showed very low cytokine responses compared to those injected with viruses.

We have written “Compared to the control group, all mice immunized with inactivated EV-A71, CV-A16, ChiE71-1-BC and ChiE71-1-EF elicited high IFN-γ, IL-2, IL-4 and IL-10 production (Figure 6E).”

The previously reported epitopes of EV-A71 and CV-A16 is addressed in lines 80-89. Figure 2B showed the differences between Ev-A71 and CV-A16 loops.

5. The authors need to explain the significance of the test on thermal stability of ChiE71. What is the relation of this assay with the immune responses induced by the chimeric EVs?

As the temperature increases, the virus capsid loss its conformation, unfold, expose the hydrophobic regions concealed in the native capsid and release its genome. This assay is used to characterize virus capsid stability and genome release for development of thermostable vaccine.  Thermal stability is one of the experiments conducted to characterize the ChiE71 as a potential vaccine candidate.

Many studies have showed that thermostable virus (high Tr) is usually attenuated , making it a good vaccine candidate. See papers:

  1. Nguyen et al., 2019. Identification and Characterization of a Poliovirus Capsid Mutant with Enhanced Thermal Stability. doi: 10.1128/JVI.01510-18
  2. Fox et al., 2017. Genetically Thermo-Stabilised, Immunogenic Poliovirus Empty Capsids; a Strategy for Non-replicating Vaccines. doi: 10.1371/journal.ppat.1006117

Many factors could change the Tr and affect infectivity and immunogenicity as showed in previous study:

  1. Increased particle thermotolerance could decreased efficiency of cell entry but increased virulence in mice (https://www.biorxiv.org/content/10.1101/2022.09.28.506941v1.full)
  2. EV-A71 infectious clone carrying conserved VP1-P157A replicated well but showed smaller plaque size, lower viral growth kinetics, and thermal instability at 39.5°C when compared to the WT. (https://www.mdpi.com/1999-4915/14/2/364)
  3. Emergence of thermally resistant EV-A71 virus with repeated passage and incubation at 52.5 °C for 30 min. (https://www.microbiologyresearch.org/content/journal/jgv/10.1099/jgv.0.001771)

6. The legends of Fig. 6 is too simple to clearly describe the results of the passive immunization (line 528). The authors should add “challenge” to the subtitle of Fig. 6B as 6D.

We would like to keep the figure legend concise, more information on the experiments can be obtained from the methodology section. We have amended the Figure 6 accordingly.

Reviewer 3 Report

The manuscripts “Substitution of coxsackievirus A16 VP1 BC and EF loop altered the protective immune responses in chimeric enterovirus A71” authors have generated Chimeric EV-A71 by substituting corresponding 7 loops by CV-A16. The manuscript is well written and in detail, I have few comments below.

1.     Provide schematic diagram of virus for easy understanding of the structures and loop of virus in Introduction.

2.     Line 90 add bracket before Fischer Scientific

3.     Provide some references of previous studies in material methods section for e.g. VN, Thermal stability etc.

4.     In general, provide the sources of monoclonal antibodies, antiserum etc throughout the manuscript.

5.      Line 211 source of HRP conjugate.

6.     Line 288, VP1 EF loop missing among VP1 BC……..and HI loop.

7.     Lines 336-337 “The ChiE71-1-EF plaque size was 25% smaller and ChiE71-1-BC showed a reduction of up to 80% in plaque size”. Could you elaborate this plaque size reduction is compared to which chimeric loop virus.

Author Response

1. Provide schematic diagram of virus for easy understanding of the structures and loop of virus in Introduction.

Thank you. We have added this as Figure 1.

2. Line 90 add bracket before Fischer Scientific

We have corrected this.

3. Provide some references of previous studies in material methods section for e.g. VN, Thermal stability etc.

We have added appropriate definition and citation.

4. In general, provide the sources of monoclonal antibodies, antiserum etc throughout the manuscript.

We have added accordingly.

5. Line 211 source of HRP conjugate.

We have added accordingly.

6. Line 288, VP1 EF loop missing among VP1 BC……..and HI loop.

We have added accordingly.

7. Lines 336-337 “The ChiE71-1-EF plaque size was 25% smaller and ChiE71-1-BC showed a reduction of up to 80% in plaque size”. Could you elaborate this plaque size reduction is compared to which chimeric loop virus.

We have edited this.

Round 2

Reviewer 1 Report

The Authors have answered all of the queries by the Reviewer. The article is suitable for publication.